# Prevalence of primary dysmenorrhea and its associated factors among adolescent girls in a rural setting of Pakistan

**Naureen Rehman** [1], **Muzna Hashmi**[1], **Arjumand Rizvi**[1], **Muhammad Sajid**[2], **Unab Khan**[3], **Saleema Gulzar**[4], **Imran Ahmed Chauhadry**[2,5], **Jai K. Das** [1,5]*

1 Institute for Global Health and Development, Aga Khan University, Karachi, Pakistan, 2 Centre of Excellence in Women and Child Health, Aga Khan University, Karachi, Pakistan, 3 Department of Family Medicine, Aga Khan University, Karachi, Pakistan, 4 School of Nursing and Midwifery, Aga Khan University, Karachi, Pakistan, 5 Department of Paediatrics and Child Health, Aga Khan University, Karachi, Pakistan

* jai.das@aku.edu

## Abstract

Primary dysmenorrhea (PD) is prevalent among adolescent girls but understudied in rural Pakistan. This study examines the prevalence of PD and its associated factors among girls aged 10–19 in a rural district of Pakistan. A cross-sectional survey was conducted in the district of Tando Mohammad Khan (TMK) among post-menarchal girls aged 10–19 using a multistage cluster random sampling. Analyses were conducted using STATA 15.0, with multivariable logistic regression adjusted for survey weights. Among 405 participants, 60.4% (95% CI [54–66]) reported PD. Higher body mass index was associated with reduced odds (overweight/obese: AOR = 0.3, 95% CI [0.1–0.9]), while lifestyle factors like physical inactivity (AOR = 1.8, 95% CI [1.2–2.9]) and inadequate sleep (AOR = 2.4, 95% CI [1.2–3.8]), increased PD odds. Dietary intake data suggested that each 10 g increase in carbohydrate slightly raised (AOR = 1.0, 95% CI [0.9–1.1]), whereas every 1 mg increase in vitamin B6 reduced PD odds (AOR = 0.9, 95% CI [0.8–0.9]). Menstrual hygiene practices like changing pads less frequently than 6 hours increased PD odds (AOR = 2.1, 95% CI [1.3–3.4]), while cloth pad reuse was associated with lower odds (AOR = 0.6, 95% CI [0.4–1.0]). PD prevalence is high among adolescents in TMK, with higher BMI linked to reduced PD odds, while diet, lifestyle, and menstrual hygiene significantly influence menstrual health. Further research is needed to guide targeted interventions for adolescent girls' well-being and to promote gender equity.

## Introduction

Menstruation marks a pivotal milestone in a young woman's life, signalling the onset of reproductive maturity and serving as an important indicator of

**Data availability statement:** The dataset required to replicate the findings of this study is provided as a Supporting file. The dataset is fully de-identified and includes all variables used in the analyses.

**Funding:** The author(s) received no specific funding for this work.

**Competing interests:** The authors have declared that no competing interests exist.

reproductive health [1]. This natural biological process often brings physical and emotional changes that can significantly affect a girl's well-being [1]. Among these, dysmenorrhea, also known as painful menstrual cramps, is one of the most common and debilitating issues faced by adolescent girls. It manifests as either primary dysmenorrhea (PD), resulting from increased prostaglandin production and stronger uterine contractions without any pelvic pathology, or secondary dysmenorrhea (SD), which stems from underlying medical conditions [1]. Both forms of dysmenorrhea can substantially impair daily functioning, contributing to school absenteeism, reduced academic performance, diminished quality of life, and heightened psychological stress during a critical period of physical and social development [2].

Globally, PD affects 45–95% of adolescents, with especially high prevalence across South Asia [3]. In Gujarat, India, 75% of girls under 20 reported for experiencing PD [4], while the figure climbs to 89% among girls aged 15–18 in Nepal [5]. In Pakistan, prevalence varies by population; for instance, PD affects 53% of girls in Mardan [6] and 78% of school-going adolescents in Karachi[2]. However, evidence from urban and school-based populations may not be generalizable to rural adolescents in Pakistan, who experience markedly different socioeconomic conditions, lower educational attainment, limited access to healthcare and menstrual health information, distinct cultural norms surrounding menstruation, and different dietary and physical activity patterns. Despite these contextual differences, adolescents living in rural parts of Pakistan remain critically understudied.

Multiple factors influence the prevalence and severity of PD. Body Mass Index (BMI) plays a key role, with both underweight and overweight adolescents at greater risk due to hormonal imbalances and inflammation [7]. South Asians are more likely to experience obesity-related health risks even at lower BMI levels (>23 kg/m²) [8]. Nutritional deficiencies, including omega-3 fatty acids [9], magnesium [10], vitamin B6 [11], calcium [10], and iron [12] may further exacerbate PD [10], particularly in rural areas where access to nutrient-rich foods is limited. Sedentary lifestyles, reduced physical activity, and poor sleep patterns may contribute to hormonal disruption [13]. In rural settings undergoing social and technological transitions, increasing screen time, changing schooling patterns, reduced recreational activity, and shifts in daily routines may alter traditional activity levels and sleep behaviours [13]. Compounding these risks are inadequate menstrual hygiene practices, influenced by deep-rooted cultural taboos, that may increase discomfort and the risk of infection [14].

Despite its high prevalence, PD remains underreported and poorly managed, particularly in rural communities. This is often attributed to cultural taboos, limited access to healthcare, and the widespread normalization of menstrual pain. There is limited research examining the condition in relation to population-specific risk factors. Therefore, this study aims to address these gaps by estimating the prevalence of primary dysmenorrhea and examining its association with sociodemographic, dietary, lifestyle, and menstrual health and hygiene factors among adolescent girls in a rural district of Sindh.

## Materials and methods

### Ethics statement

Ethical approval for the primary study was secured from the Aga Khan University's Ethical Review Committee (2021-5943-16892). For minors aged less than 18 years, written consent was obtained from a legal guardian, and written assent was obtained from the participant. For participants aged 18–19 years, written informed consent was obtained directly. All participants were informed of their right to refuse participation or withdraw from the study at any time without prejudice. For this study analysis, ethical review exemption was granted by The Aga Khan University Ethical Review Committee (2024-9948-28820).

### Objective

- To estimate the prevalence of PD among adolescent girls of 10–19 years in rural Sindh, Pakistan.

- To assess the relationship between sociodemographic, dietary, lifestyle, and menstrual health and hygiene factors with PD among adolescent girls of 10–19 years in rural Sindh, Pakistan.

### Study design

The Adolescent Health and Well-being Survey (AD-HAWS) was conducted in the year 2021 in Tando Muhammad Khan (TMK) district, Sindh, primarily aimed to investigate the dietary intake and its associations with anthropometric measurements and health outcomes among adolescents aged 10–19, both in and out of school. This study estimates the prevalence of PD and explores its relationship with sociodemographic, family, nutritional, lifestyle, menstrual health and hygiene factors. Reporting adheres to STROBE guidelines [15].

### Study setting

The study took place in TMK, a rural district in Sindh, Pakistan, spanning 1814 km² with 677,228 residents (51% females). TMK has three talukas, 17 Union Councils, a population density of 75.8/km², and a 2.3% annual growth rate. Poverty affects 89.3% of the population, resulting in limited access to education and employment. Around 62% of children are out of school, and the education budget allocates only 15% for girls compared to 48% for boys [16].

### Study population and eligibility criteria

Participants were adolescent girls aged 10–19 years who were permanent residents of TMK and had attained menarche. Those who were pregnant, had chronic medical conditions, or were taking specific medications were excluded. Age verification was conducted through official documents such as birth certificates, B-forms, vaccination cards, or school records, or through caregiver recall. Only one adolescent was selected per household, which was defined by the sharing of food and shelter. Informed consent was obtained from guardians for participants aged 10 to under 18 years, while those aged 18–19 provided direct consent in the local language.

### Sample size

The study sample size was calculated to estimate the prevalence of underweight among adolescents aged 10–19 years with a 95% confidence level, 5% precision, a design effect of 2, and an expected response rate of 90%. Using baseline data from the National Nutrition Survey (NNS 2018) [17], which reported underweight prevalence of 30.7% in boys and 20.1% in girls, a total of 1304 households were selected for the adolescent survey, with an equal gender distribution (652 males and 652 females). Of these, 405 adolescent girls were eligible to participate in the study.

## Sampling strategy

The sampling strategy employed multistage cluster sampling, with the primary sampling units (PSUs) comprising 48 rural clusters of the COMIC trial [18]. All households within the selected clusters were listed to create a comprehensive line listing, which served as the sampling frame for final household selection. Systematic sampling was then applied, where every kth household was selected after a random start, with k calculated based on the total number of households and the required sample size per cluster. Within each selected household, if multiple adolescents were eligible, one was chosen using the Kish grid method, which assigns random numbers to individuals based on their listing order. Households that refused to participate were replaced by the next household.

## Study questionnaire and pilot testing

The survey tools were finalized after thorough review by team members, collaborators, and external experts, translated into Urdu, and back-translated to ensure accuracy. Pilot testing with up to 50 participants assessed response latency, interpretation, and appropriateness. Validity was reviewed by public health and nutritional epidemiology experts, and feedback guided refinements.

## Study variables

**Outcome.** The outcome variable was self-reported menstrual pain, used as a proxy for PD. Menstrual health and hygiene, assessed in Module O of the study questionnaire, included the item "Do you have pain during menstruation?" adapted from the Menstrual Health Questionnaire (MHQ) [19] and Menstrual Practices Questionnaire (MPQ) [20]. No clinical examination or diagnostic screening was conducted to exclude underlying gynecological conditions; therefore, classification was based solely on participant self-report.

**Covariates.** Anthropometric assessments of adolescents were conducted at the household level by trained staff following standard methods. Weight and height were measured to the nearest 0.1 kg and 0.1 cm using calibrated Seca instruments (floor scale model 813, stadiometer model 213), with participants in light clothing and without shoes. Mid-Upper Arm Circumference (MUAC) was measured with a Seca tape (model 201). All measurements were taken in duplicate, and discrepancies exceeding 1 cm for height or 0.5 kg for weight were resolved through a third measurement by the team leader. BMI was calculated as weight (kg) divided by height squared (m²). This study adopted South Asian BMI ranges, classifying underweight as less than 18.5 kg/m², normal weight as 18.5–22.9 kg/m², and overweight as 23 kg/m² and above [21].

## Data collection tools

This study utilized validated tools to assess nutrition and lifestyle factors. Household food insecurity was measured using the FAO Global Food Insecurity Experience Scale [22], dietary diversity with the USAID FANTA Household Dietary Diversity Scale [23], and meal patterns using tools from epidemiological studies [24,25]. The WHO Health Behaviour in School-aged Children (HBSC) [26] and Global School-based Student Health Survey (GSHS) tools [27] assessed lifestyle, physical activity, sleep patterns, and tobacco use, while depression was evaluated via Beck's Depression Inventory [28]. Sleep duration was assessed through self-reported questions capturing both daytime and nighttime sleep separately for weekdays and weekends; average daily sleep duration was calculated by summing reported daytime and nighttime sleep and computing a weekly average across all seven days. Adequate sleep was defined as an average duration of 6–8 hours per night. Physical activity was assessed using self-reported recall of moderate-to-vigorous activity for each day of the week, with participants reporting duration separately for weekdays and weekends; total weekly minutes were calculated by summing daily durations across all seven days. Physical activity was classified as adequate if adolescents reported engaging in ≥180 minutes of moderate-to-vigorous activity per week.

Nutrient intake, including macronutrients and micronutrients, was analyzed through a 24-hour dietary recall. Nutrient intake was calculated by multiplying the frequency weight, portion size, and nutrient content for each food item. Food composition data were sourced from the MAL-ED database [29], USDA National Nutrient Database [30], and Food Composition Table for Bangladesh [31]. A trained nutritionist assigned food codes and applied retention factors to account for nutrient losses due to cooking. Total energy, macronutrient, and micronutrient intakes were computed for each participant and incorporated into the regression models.

### Data collection and participant recruitment

For the primary study, recruitment of study participants was conducted from 01/03/2021–31/05/2021. Household-level information including demographic and socioeconomic indicators, as well as health outcomes was obtained from the mother or caregiver after obtaining informed consent. Data were collected by trained enumerators using handheld devices (Samsung tablets running Android 5.1), following six days of training on content, operational procedures, and data management. To minimize errors, the data entry program incorporated range checks, consistency checks, and skip patterns. Data were synced daily and uploaded from the field sites to the university server. The data management unit generated daily summary reports for quality assurance, which were shared with field teams for rectification when necessary. All data were encrypted, securely stored, and fully anonymize. For this study, we utilized the already available data.

### Statistical analysis

All statistical analyses were performed using STATA 15.0. Descriptive statistics included means and standard deviations for continuous variables, and categorical variables were reported as weighted percentages. Missing data were assessed for a Missing Completely at Random (MCAR) pattern. Of the 405 eligible adolescent girls included in the analysis, missing values were minimal, with 10 participants (2.5%) having incomplete data for nutrient intake variables. No participants were excluded due to missingness and multiple imputation was performed incorporating sociodemographic, household, dietary, lifestyle, and outcomes [32].

Sample weights adjusted for multistage cluster sampling were incorporated into the analysis for sociodemographic, household, family, dietary, lifestyle, and menstrual health factors. Weighted multiple logistic regression was used to evaluate associations between covariates and PD, with multicollinearity assessed. Given the exploratory nature of identifying potential determinants in an understudied rural population, a significance threshold of $p < 0.10$ was adopted in the multivariable model to reduce the risk of Type II error. Adjusted odds ratios (AOR) with 95% confidence intervals (CIs) were reported to facilitate interpretation of effect sizes. As the analysis utilized a pre-existing dataset (S1 Data), a post hoc power analysis was performed to evaluate whether the available sample size was sufficient to detect statistically significant differences in the outcome based on observed effect sizes.

### Results

A total of 611 girls were included in the survey, of which 405 girls were eligible to be included in the final analysis (Fig 1).

The mean age of participants was $15.5 \pm 0.09$ years, with nearly one-third (31.9%) aged below 15. In terms of education, most participants had no formal schooling (66.4%). Additionally, parental education levels revealed that a large majority of mothers were uneducated (85.6%), compared to 56.7% of fathers. In addition, household food insecurity was prevalent, with 76.5% of participants living in moderately to severely food-insecure households. Despite this, dietary diversity was generally high, with 92.8% reporting access to a varied diet. Furthermore, 24% of participants were stunted, 60.2% were underweight, and 4.5% were overweight. Daily nutrient intake revealed average consumption of 245.7 g of carbohydrates, 33.0 g of protein, and 36.7 g of fat. Micronutrient intake included 158.3 mcg of vitamin A, 2.7 mcg of vitamin B6, 31.0 mg of vitamin C, and 0.5 mcg of vitamin D. Mean calcium intake was 321.6 mg, iron was 6.2 mg, zinc was 5.0 mg, and folate was 117.5 mcg (Table 1).

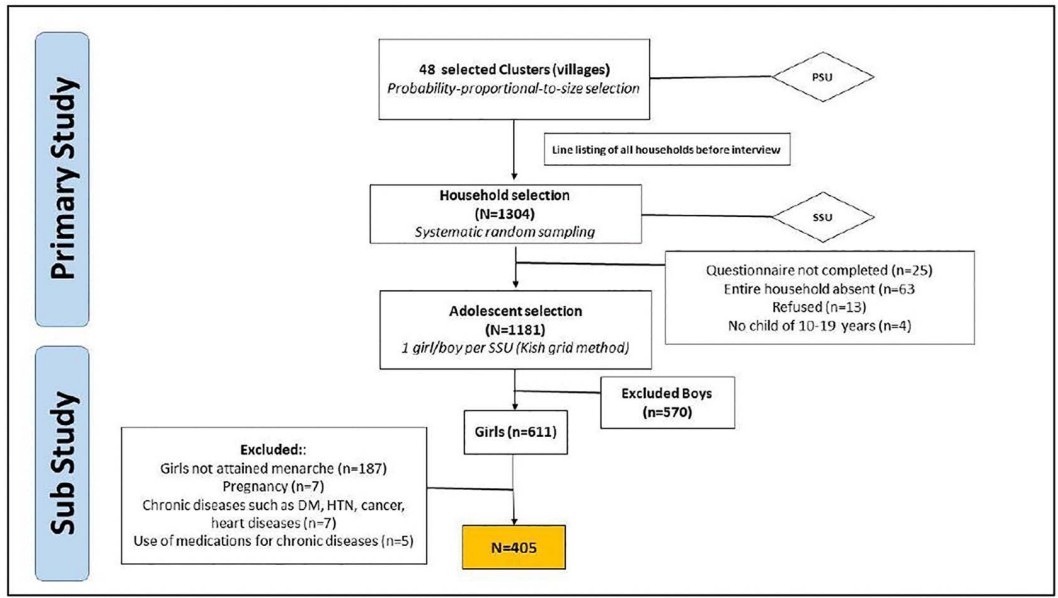

**Fig 1. Hierarchical Sampling Strategy involving multi-stage cluster random sampling.**

The overall prevalence of PD among adolescent girls was 60.4% (95% CI [54–66]). Additionally, 53.2% reported a family history of PD, suggesting a potential hereditary link. Most participants experienced menarche after the age of 12 (70.1%), while only 4.5% had irregular menstrual cycles. Regarding menstrual hygiene practices, 91.2% used cloth pads, with 67.2% reporting its reuse. Moreover, 92.8% changed their sanitary materials after more than six hours, and 81.2% bathed regularly during menstruation (Table 2).

In univariate logistic analysis, the odds of PD were higher among girls older than 15 years (COR = 1.5, 95% CI [0.9–2.4]), those from high-income households (COR = 1.5, 95% CI [0.9–2.4]), and those with access to improved toilet facilities (COR = 1.7, 95% CI [1.1–2.8]). Girls who were middle (COR = 1.1, 95% CI [0.6–1.9]) or eldest children in the family (COR = 1.8, 95% CI [0.9–3.3]) also showed increased odds of PD compared to those who were youngest children. Overweight/obese girls had lower odds of PD (COR = 0.2, 95% CI [0.1–0.7]) compared to those with normal BMI. Nutrient intake analysis showed that with every 10 g increase in carbohydrate intake, the odds of PD increased slightly (COR = 1.02, 95% CI [0.9–1.04]), while every 1 mg increase in vitamin B6 intake was associated with reduced odds of PD (OR = 0.9, 95% CI [0.8–0.9]). Girls with inadequate sleep had higher odds of PD (COR = 1.7, 95% CI [0.9–3.1]), and those with low physical activity were more likely to report PD (COR = 2.1, 95% CI [1.3–3.2]). A strong association was observed with family history of PD (COR = 22.7, 95% CI [13.1–37.2]), and girls who experienced menarche after age 12 had increased odds (COR = 1.9, 95% CI [1.2–3.0]). Menstrual hygiene practices also showed significant associations, with higher odds of PD among girls who changed sanitary products after more than six hours (COR = 2.3, 95% CI [1.5–3.6]) and lower odds of PD among those who reused cloth pads (COR = 0.7, 95% CI [0.4–1.1])(Table 3).

Multivariable analysis revealed that girls who were middle children (AOR = 1.7, 95% CI [0.8–3.4]) had higher odds of PD compared to the youngest children, while eldest-child status (AOR = 1.0, 95% CI [0.5–1.8]) showed no significant association. Overweight or obese girls (AOR = 0.3, 95% CI [0.1–0.9]) were less likely to experience PD. Each 10 grams increase in carbohydrate intake (AOR = 1.0, 95% CI [0.9–1.1]) corresponded to a slight, significant rise in PD odds, whereas higher vitamin B6 intake (AOR = 0.9 per 1 mg increase, 95% CI [0.8–0.9]) was significantly associated with lower odds. Girls engaging in less than 180 minutes of physical activity per week (AOR = 1.8, 95% CI [1.2–2.9]) and those reporting sleep

**Table 1. Sociodemographic and nutritional characterstics of adolescent girls aged 10-19 years in TMK (N = 405).**

| Sociodemographic and nutritional characteristics | n(%) |
|---|---|
| Sociodemographic characteristics | |
| Age group (Years) | |
| <15 | 129(31.9) |
| ≥15 | 276(68.1) |
| Educational level | |
| None | 269(66.4) |
| Primary | 83(20.5) |
| Secondary | 28(6.9) |
| High school or more | 25(6.2) |
| Household food insecurity status | |
| Mild food insecurity | 73(20.9) |
| Moderate to severe insecurity | 264(76.5) |
| Food secure | 9(2.6) |
| Household dietary diversity | |
| Lowest | 2(0.5) |
| Medium | 27(66.7) |
| Highest | 376(92.8) |
| *Nutritional characteristics | |
| Mean Nutrients intake per day | |
| Carbohydrates (g) | 245.7(73.4) |
| Proteins (g) | 33.0(11.9) |
| Fat (g) | 36.7(24.8) |
| Vitamin A (mcg) | 158.3(118.6) |
| Vitamin B6 (mcg) | 2.7(9.8) |
| Vitamin C (mg) | 31.0(9.1) |
| Vitamin D (D2 + D3) (mcg) | 0.5(0.1) |
| Calcium (mg) | 321.6(182.0) |
| Iron (mg) | 6.2(1.9) |
| Zinc (mg) | 5.0(1.9) |
| Folate (mcg) | 117.5(94.1) |

*mean (SD)

durations below six or above nine hours (AOR = 2.4, 95% CI [1.2–3.8]) were more likely to have PD. Menarche after 12 years of age (AOR = 1.3, 95% CI [0.7–2.2]) showed a non-significant increase in PD risk. Menstrual hygiene also played a notable role, as girls who changed sanitary pads after more than 6 hours (AOR = 2.1, 95% CI [1.3–3.4]) had markedly higher odds of PD. However, the girls who reused their cloth pads (AOR = 0.6, 95% CI [0.4–1.0]) had significantly less chance of experiencing PD as compared to their counterparts.

A post hoc power analysis was conducted to assess the adequacy of the sample for detecting observed effect sizes in the multivariable model. Results indicated that the model achieved greater than 80% power for detecting associations where the odds ratios were less than 0.6 or greater than 1.6. For predictors with odds ratios closer to the null (between 0.6 and 1.6), the statistical power was comparatively lower, indicating that non-significant findings in this range may reflect insufficient sensitivity to detect modest effects.

**Table 2. Menstrual health and hygiene practices of adolescent girls aged 10-19 years in TMK (N = 405).**

| Menstrual Health and Hygiene practices | n(%) |
|---|---|
| Primary dysmenorrhea | 247(60.4) |
| Family History of Primary dysmenorrhea | 221(53.2) |
| Age at menarche (years) >12 | 287(70.1) |
| Irregular cycle pattern | 18 (4.5) |
| Use of cloth pads | 371(91.2) |
| Reusing cloth pads | 271(67.2) |
| Changing sanitary pads >6 hours | 376(92.8) |
| Regular bathing during menstruation | 329(81.2) |

## Discussion

The study suggested that the prevalence of PD among 10–19-year-old girls in the rural district of Pakistan was 60.4%. This prevalence can be considered relatively low compared to similar studies conducted in regional countries, where prevalence rates are often reported higher. For instance, studies in neighbouring countries such as Iran (85%) [33], India (75%) [4] and Turkey (72%) [34]. It should be noted, however, that differences in measurement tools, age ranges, and study populations may partly account for these variations, which warrants cautious interpretation. The lower prevalence in our study may reflect underreporting, especially in rural areas where cultural norms and limited awareness lead adolescents to view PD as a normal part of menstruation.

Our study also examined sociodemographic, nutritional, lifestyle, and menstrual health and hygiene factors associated with PD. Middle-born children, insufficient physical activity, inadequate sleep, and delayed changing of sanitary pads were significant risk factors. Higher carbohydrate intake was associated with a marginally significant increase in PD odds, whereas menarche after 12 years of age was not significantly associated with PD. In contrast, overweight/obesity, higher vitamin B6 intake, and cloth pad reuse were associated with lower odds of PD.

The study showed that middle-born children had a significantly higher likelihood of experiencing PD compared to the youngest children in the family. One possible explanation may relate to sociocultural dynamics, as middle-born girls in some contexts may receive less attention or support, particularly when siblings differ by gender [35]. It is plausible that such dynamics could contribute to emotional stress among middle daughters, which in turn may influence pain perception [35]. However, these interpretations are speculative, and further research is needed to determine whether family structure and birth order influence dysmenorrhea through psychosocial pathways.

Surprisingly, our findings indicate that higher BMI (overweight and obese) was associated with lower odds of PD. These results may align with existing literature suggesting that overweight and obese girls may experience less PD due to biological mechanisms, such as increased adipose tissue providing cushioning during menstruation, which can help alleviate menstrual pain [7,36,37]. However, this study contradicts most of the literature that highlights higher BMI is generally associated with greater levels of menstrual pain [38,39]. This discrepancy may be attributed to the fact that many South Asian studies have not utilized region-specific BMI ranges. Furthermore, a study from Japan highlighted that individuals experiencing greater PD severity had lower intakes of essential nutrients like animal proteins, vitamin D, and vitamin B12 [40]. This suggests that overweight individuals may have dietary habits that contribute to better nutritional intake, potentially normalizing estrogen levels and thereby mitigating menstrual pain [40–42]. However, these explanations are hypothetical, and further longitudinal and mechanistic studies are needed to clarify the relationship between BMI, diet, and PD.

Our study found that higher carbohydrate intake showed a marginally significant association with increased odds of PD. Existing literature suggests that high carbohydrate consumption, particularly from refined sources, may contribute to

**Table 3. Determinants of PD among adolescent girls aged 10-19 years in TMK (N = 405).**

| Characteristics | Total (n) | PD % | COR (95% CI) | AOR (95% CI) |
|---|---|---|---|---|
| **Sociodemographic Characteristics** | | | | |
| Age groups (years) | | | | |
| ≤15 | 129 | 53.5% | Ref. | |
| >15 | 276 | 63.7% | 1.5 (0.9-2.4)* | |
| Attended school | | | | |
| No | 269 | 61.6% | 1.1 (0.7-1.8) | |
| Yes | 136 | 58.1% | Ref | |
| Marital status | | | | |
| Married | 42 | 68.9% | Ref | |
| Single | 363 | 59.5% | 0.6 (0.3-1.3) | |
| Wealth status (terciles) | | | | |
| Lowest | 136 | 55.3% | Ref | |
| Middle | 134 | 61% | 1.2 (0.7-2.0) | |
| Highest | 135 | 65.7% | 1.5 (0.9-2.4)* | |
| Toilet facilities | | | | |
| Unimproved | 254 | 55.6% | Ref | |
| Improved | 151 | 69.1% | 1.7 (1.1-2.8)* | |
| **Family characteristics** | | | | |
| Mother education | | | | |
| No Education | 347 | 60.5% | 1.0 (0.5-1.8) | |
| Some Education | 58 | 59.7% | Ref | |
| Father education | | | | |
| No Education | 227 | 58.7% | 0.8 (0.5-1.2) | |
| Some Education | 178 | 63% | Ref | |
| Birth order | | | | |
| Eldest | 223 | 58.3% | 1.8 (0.9-3.3)* | 1.0 (0.5-1.8) |
| Middle | 107 | 68.8% | 1.1 (0.6-1.9)** | 1.7 (0.8-3.4)* |
| Youngest | 75 | 54.8% | Ref | Ref |
| **Dietary and nutritional characteristics** | | | | |
| BMI | | | | |
| Underweight (<18.5) | 246 | 58.9% | 0.7 (0.4-1.2)** | 0.6 (0.3-1.2)* |
| Normal (18.5-22.9) | 141 | 65.6% | Ref | Ref |
| Overweight/Obese (>22.9) | 18 | 34.7% | 0.2 (0.1-0.7)* | 0.3 (0.1-0.9)* |
| Stunting | | | | |
| Stunted | 97 | 60.7% | 1.0 (0.6-1.6) | |
| Normal | 308 | 61.3% | Ref | |
| **Nutrients intake per day (mean(SD))** | | | | |
| Carbohydrates | – | 250.8(79.8) | 1.02 (0.9-1.04)* | 1.0 (0.9-1.1)* |
| Proteins | – | 33.5(11.7) | 1.0 (0.9-1.0) | |
| Fats | – | 36.0(23.4) | 0.9 (0.9-1.) | |
| Vitamin B6 | – | 152.8(101.8) | 0.9 (0.8-0.9)* | 0.9 (0.8-0.9)* |
| Folate | – | 89.9(96.3) | 0.9 (0.9-1.0) | |
| Iron | – | 6.8(1.5) | 1.0 (0.9-1.1) | |

*(Continued)*

**Table 3.** (Continued)

| Characteristics | Total (n) | PD % | COR (95% CI) | AOR (95% CI) |
|---|---|---|---|---|
| **Lifestyle characteristics** | | | | |
| Screen time | | | | |
| ≤3 hours | 227 | 59.5% | Ref | |
| >3 hours | 178 | 61.6% | 1.1 (0.7-1.7) | |
| Physical activity | | | | |
| Inactive (<180 mins/week) | 211 | 69.2% | 2.1 (1.3-3.2)* | 1.8 (1.2-2.9)* |
| Active (≥180 mins/week) | 194 | 51.6% | Ref | Ref |
| Smokeless tobacco use | | | | |
| Yes | 73 | 58.3% | Ref | |
| No | 332 | 60.9% | 0.9 (0.5-1.6) | |
| Sleep patterns | | | | |
| Adequate (6–9 hours/day) | 301 | 56.2% | Ref | Ref |
| Inadequate (<6 or >9 hours/day) | 104 | 71.6% | 1.7 (0.9-3.1)* | 2.4 (1.2-3.8)* |
| **Menstrual health factors** | | | | |
| Age at menarche (years) | | | | |
| ≤ 12 years | 118 | 48.7% | Ref | Ref |
| >12 years | 287 | 65.3% | 1.9 (1.2-3.0)* | 1.3 (0.7-2.2) |
| Cycle pattern | | | | |
| Regular | 387 | 60.3% | Ref | |
| Irregular | 18 | 63.7% | 0.9 (0.3-2.6) | |
| Menstrual cycle duration | | | | |
| 3-7 days | 355 | 58.9% | Ref | |
| <3 or>7 days | 50 | 70.5% | 1.5 (0.7-3.0) | |
| Family history of PD | | | | |
| Yes | 221 | 89.2% | 22.7 (13.1-37.2)* | |
| No | 184 | 26.3% | Ref | |
| **Menstrual hygiene practices** | | | | |
| Type of Sanitary product use | | | | |
| Sanitary napkin | 34 | 51.5% | Ref | |
| Cloth | 371 | 61.4% | 1.5 (0.6-3.6) | |
| Frequency of changing sanitary product | | | | |
| ≤6 hourly | 151 | 47.1% | Ref | Ref |
| > 6 hourly | 254 | 67.6% | 2.3 (1.5-3.6)* | 2.1 (1.3-3.4)* |
| Reusing cloth pad | | | | |
| No | 100 | 67.1% | Ref | Ref |
| Yes | 271 | 59.3% | 0.7 (0.4-1.1)** | 0.6 (0.4-1.0)* |
| Regular bathing during periods | | | | |
| No | 294 | 59.6% | Ref | |
| Yes | 111 | 62.5% | 1.1 (0.7-1.8) | |

**p<0.25,*p<0.1

Abbreviations: CI: confidence interval; COR: crude odds ratio; AOR: Adjusted odds ratio

insulin resistance and elevated inflammation, both of which have been implicated in worsening menstrual pain. In contrast, vitamin B6 intake significantly reduced the odds of PD. Vitamin B6, abundant in foods such as poultry, fish, potatoes, and bananas, is essential for synthesizing neurotransmitters that regulate mood and pain perception [43]. It may also lower prostaglandin levels, thereby reducing menstrual pain severity [43]. Furthermore, vitamin B6 enhances magnesium absorption and synergizes with omega-3 fatty acids, strengthening their anti-inflammatory effects and further alleviating menstrual discomfort [43].

Having a physically active lifestyle has been shown to protect against PD by improving blood circulation, reducing inflammation, and regulating hormonal levels through the balance of estrogen and progesterone, which lowers prostaglandin production and eases menstrual cramps [44]. In our study, physically inactive girls had higher odds of experiencing PD, consistent with evidence supporting the benefits of regular exercise in alleviating menstrual pain. For instance, a study in Iran reported that adolescents who exercised regularly were significantly less likely to report severe PD ($\beta = -0.17$, $p < 0.001$) [45]. However, the increasing prevalence of physical inactivity among adolescents, driven by sedentary behaviours such as excessive screen time, raises concerns for menstrual health.

Inadequate sleep emerged as a strong risk factor for poor menstrual health. This aligns with findings from a study in Lahore, Pakistan, where poor sleep quality was significantly associated with increased menstrual pain and dysmenorrhea among adolescent girls. Insufficient sleep can disrupt the regulation of hormones such as cortisol and melatonin, thereby affecting pain perception and menstrual health [46]. In rural areas of Pakistan, socioeconomic challenges, low living standards, and cultural norms often limit access to healthcare and education, further exacerbating sleep deprivation due to household responsibilities and social expectations [47]. This cycle of inadequate sleep and heightened menstrual discomfort highlights the need for awareness and targeted interventions for this population.

In addition, this study found that menarche after 12 years of age was associated with a non-significant increase in the odds of PD. This contrasts with literature linking early menarche to higher risks of health issues, including PD, particularly in developed countries [48]. While early menarche is often associated with hormonal changes that elevate PD risk, several studies from developing countries, such as China [49] and Bangladesh [50], report a rising trend toward later menarche. Research from Nigeria also noted a 32% prevalence of PD among those with menarche at 13–14 years, compared to 22% among those with earlier menarche [51].

Moreover, changing pads less frequently were strongly associated with PD, likely due to inflammation from moisture retention, compounded by limited sanitation access and cultural stigma surrounding menstrual hygiene [52]. Interestingly, participants who reused clean cotton pads reported reduced pain and greater comfort, possibly due to the breathability and natural composition of cotton, which may inhibit bacterial growth [53]. Washing softens the material, making it less abrasive, while sunlight drying further improves hygiene by reducing bacterial load [53]. Sustainable menstrual products, such as cotton pads, may enhance comfort during menstruation [53]. Future qualitative research is needed to explore the experiences of women using different menstrual hygiene methods and their impact on pain and comfort.

To the best of our knowledge, this study is the first to assess factors associated with primary dysmenorrhea among adolescents in TMK. Its strengths include a robust cluster random sampling approach in rural Sindh, which enhances generalizability to the target population, and inclusion of both in- and out-of-school girls. Furthermore, employing a South Asian BMI range, unlike most studies using global standards, increases relevance to the local population. Importantly, the post hoc power for key associations ranged from 84% to 98% to detect an odds ratio above 0.3, indicating sufficient sample size to detect meaningful differences. However, several limitations should be acknowledged; primary dysmenorrhea was assessed using a single binary (yes/no) self-reported question regarding the presence of menstrual pain, while this approach is commonly used in large-scale surveys and facilitates feasibility in field-based data collection, it does not capture the pain severity, duration, frequency, or associated functional impairment, which may lead to an oversimplification of symptom experiences. In addition, the limited categorization of age and the cross-sectional study design restrict causal

inference. Potential underreporting related to cultural stigma surrounding menstruation and logistical challenges in data collection, particularly in rural settings, further warrants cautious interpretation of the findings. Sleep duration and physical activity were assessed using self-reported seven-day recall and may be subject to recall or social desirability bias. Future studies with larger samples should explore whether the association between dietary intake and PD varies across BMI categories to assess potential effect modification.

This research has significant implications for public health policies and educational programs targeting menstrual health. School health programs should educate adolescents about menstrual health, emphasizing the importance of nutrition, hygiene practices, and symptom recognition. Additionally, parent-focused campaigns can foster supportive environments that encourage open discussions about menstrual health, enhancing the reporting and management of PD. Given that a majority of girls are out of school, community-wide initiatives involving various stakeholders, such as local healthcare providers, NGOs, and community leaders, are essential. Raising awareness about dietary intake and its influence on PD is crucial. Interventions could be delivered through community health workers, mobile clinics, or other community-based platforms to ensure broad reach and accessibility. These initiatives should aim to create awareness, provide education, and distribute resources related to menstrual health. Such efforts align with Sustainable Development Goal 5, which focuses on achieving gender equality and empowering all women and girls.

## Conclusion

This study highlights that overweight status is linked to lower odds of PD among adolescent girls in rural Pakistan, while factors like higher carbohydrate intake, inadequate sleep, physical inactivity, late menarche, and inadequate menstrual hygiene practices increase the odds of PD. Public health interventions should prioritize promoting nutrition, physical activity, menstrual hygiene education, and access to reusable pads. Future research should explore longitudinal effects of these factors on PD to better inform targeted strategies. These efforts align with SDG 5, aiming to empower girls and enhance their well-being.

## Supporting information

**S1 Data. Study dataset used for the analysis.**
(XLS)

## Acknowledgments

This work was part of the Naureen Rehman thesis for MSc in EpiBio from Aga Khan University. We gratefully acknowledge Sir Iqbal Azam, Assistant Professor in the Department of Community Health Sciences, for his invaluable support and thoughtful guidance throughout this work.

## Author contributions

**Conceptualization:** Naureen Rehman, Unab Khan, Saleema Gulzar, Jai K. Das.

**Data curation:** Muhammad Sajid, Imran Ahmed Chauhadry.

**Formal analysis:** Naureen Rehman, Muzna Hashmi, Arjumand Rizvi, Jai K. Das.

**Project administration:** Naureen Rehman, Muzna Hashmi, Jai K. Das.

**Software:** Naureen Rehman, Muzna Hashmi, Arjumand Rizvi.

**Supervision:** Jai K. Das.

**Writing – original draft:** Naureen Rehman, Muzna Hashmi.

**Writing – review & editing:** Arjumand Rizvi, Jai K. Das.

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
