## [Decision Letter · Decision Letter 0]

17 Dec 2025

PGPH-D-25-03086

Prevalence of Primary Dysmenorrhea and its associated factors among adolescent girls in rural setting of Pakistan

Dear Dr. Das,

Thank you for submitting your manuscript to PLOS Global Public Health. After careful consideration, we feel that it has merit but does not fully meet PLOS Global Public Health’s publication criteria as it currently stands. Therefore, we invite you to submit a revised version of the manuscript that addresses the points raised during the review process.

The manuscript has been evaluated by two reviewers, and their comments are available below.

The reviewers have raised a number of concerns that need attention. They request additional information on methodological aspects of the study, improvements to the Discussion and limitations of the study design, and the removal of any speculative results.

Could you please revise the manuscript to carefully address the concerns raised?

We look forward to receiving your revised manuscript.

Kind regards,

Helen Howard

Staff Editor

Journal Requirements:

1. Please ensure that your Ethics Statement is available in its entirety at the beginning of your Methods section, under a subheading 'Ethics Statement'.

2. Please upload separate figure files in .tif or .eps format. Also, remove the figures from your manuscript file but keep the legends.

3. In the online submission form, you indicated that “Data will be available upon reasonable request”.

3. Uploaded as supplementary information.

Additional Editor Comments (if provided):

Reviewers' comments:

Reviewer's Responses to Questions

**Comments to the Author**

1. Does this manuscript meet PLOS Global Public Health’s publication criteria ? Is the manuscript technically sound, and do the data support the conclusions? The manuscript must describe methodologically and ethically rigorous research with conclusions that are appropriately drawn based on the data presented.

Reviewer #1: Partly

Reviewer #2: Yes

2. Has the statistical analysis been performed appropriately and rigorously?

Reviewer #1: Yes

Reviewer #2: Yes

3. Have the authors made all data underlying the findings in their manuscript fully available (please refer to the Data Availability Statement at the start of the manuscript PDF file)?

Reviewer #1: No

Reviewer #2: Yes

4. Is the manuscript presented in an intelligible fashion and written in standard English?

Reviewer #1: Yes

Reviewer #2: Yes

Reviewer #1: The manuscript addresses an important and understudied topic—primary dysmenorrhea (PD) among rural adolescent girls in Pakistan—and has several strengths, including a robust sampling strategy and the use of region-specific BMI criteria. However, significant methodological and interpretational limitations currently prevent it from being accepted in its current form.

The abstract and results section could be more concise and focused on the most salient findings.

In the Introduction, You have mentioned ”Nutritional deficiencies, including omega-3 fatty acids, magnesium, vitamin B6, calcium, and iron may further exacerbate PD”. Then you have cited one reference. I suggest that to use the references for each nutritional items that you have listed.

For example for omega-3 fatty acid cite this article: Effect of Clupeonella grimmi (anchovy/kilka) fish oil on dysmenorrhea https://doi.org/10.26719/2010.16.4.408.

The outcome variable (PD) was assessed using a single yes/no question: “Do you have pain during menstruation?”This is a major limitation, as it does not differentiate between primary and secondary dysmenorrhea, assess pain severity, frequency, or functional impact. The authors should clearly acknowledge this as a limitation and consider validating the measure against established tools (e.g., Visual Analog Scale, Menstrual Symptom Questionnaire).

BMI and PD: The finding that overweight/obese girls had lower odds of PD contradicts much of the literature. While the authors propose possible biological and dietary mechanisms, these are speculative and not directly tested.

Categorization and Measurement of Key Variables: Menarche Age: Categorized only as ≤12 vs. >12 years, which may obscure nuanced age-related risk patterns.

Sleep and Physical Activity: Definitions (e.g., “inadequate sleep”) are not clearly justified or referenced. Please provide clearer operational definitions and, where feasible, consider more granular categorizations.

The discussion of middle-born children and emotional stress is speculative and lacks support from the study's qualitative or quantitative data. Either remove these speculative interpretations or frame them as hypotheses for future research.

Some confidence intervals cross 1.0 (e.g., middle-born AOR = 1.7 [0.8–3.4]), yet are described as “significant” or “more likely.” Use more cautious language and distinguish between statistical and clinical significance.

Some tables (e.g., Table 1) are overly detailed and could be streamlined or moved to supplementary materials. Do not repeat the results in the text and tables.

The discussion could be better organized to prioritize the most robust and actionable findings.

Reviewer #2: This is a well-conducted cross-sectional study addressing an important and under-researched topic in a rural Pakistani setting. The manuscript is clearly written, methods are generally robust, and the findings contribute meaningfully to the literature on menstrual health in low-resource contexts. I recommend acceptance after minor revisions addressing the points below.

Comments

Methods

Measurement of dysmenorrhea:The outcome is assessed with a single yes/no question (“Do you have pain during menstruation?”). This may not capture severity, frequency, or functional impact. Please justify this approach or acknowledge it as a limitation. If possible, consider referencing whether pain intensity or days affected were recorded, even if not used in analysis.

Sampling: It is stated that 405 adolescent girls were eligible from the original sample. Please clarify how many were approached vs. participated, and mention any exclusion due to missing data.

Sleep and physical activity measures: Briefly describe how these were defined/collected (e.g., self-report, recall period).

Results

Table 3: Some CIs cross 1 but are marked with asterisks (e.g., eldest child, carbohydrate intake). Please ensure footnote clearly explains that asterisks indicate p<0.1 or p<0.2, not statistical significance at 0.05.

Family history of PD shows a very high OR (~22). Please check for possible over-adjustment or collinearity in the model.

Discussion

The comparison with prevalence in Iran (85%), India (75%), etc., is useful, but note differences in measurement tools and age ranges where possible.

The public health implications are well laid out, but consider adding a sentence on how interventions could be delivered given the high out-of-school rate (e.g., community health workers, mobile clinics).

Language & Clarity

Minor grammatical edits needed (e.g., “post-menarchael” → “post-menarcheal”; “infrequent pad changing (>6 hours)” could be phrased more clearly as “changing pads less frequently than every 6 hours”).

Ensure consistency in reporting confidence intervals: sometimes written as [54–66], elsewhere as [54% to 66%].

(what does this mean? ). If published, this will include your full peer review and any attached files.). If published, this will include your full peer review and any attached files.

**Do you want your identity to be public for this peer review?** For information about this choice, including consent withdrawal, please see our Privacy Policy .

Reviewer #1: No

Reviewer #2: **Yes:** Zohreh KarimiankakolakiZohreh Karimiankakolaki

---

## [Decision Letter · Decision Letter 1]

8 Jan 2026

PGPH-D-25-03086R1

Prevalence of Primary Dysmenorrhea and its associated factors among adolescent girls in rural setting of Pakistan

Dear Dr. Das,

Thank you for submitting your manuscript to PLOS Global Public Health. After careful consideration, we feel that it has merit but does not fully meet PLOS Global Public Health’s publication criteria as it currently stands. Therefore, we invite you to submit a revised version of the manuscript that addresses the points raised during the review process.

Please carefully check your references, as there appear to be some errors. As an example, in your Tracked Changes revised manuscript, reference 11 is cited in support of a lack of vitamin B6 being associated with PD; however, reference 11 is listed as the STROBE checklist (Alkhaqani, 2022), which is likely incorrect. This is true of other references in your References list too, so people carefully check all in-text citations and their associated references.

Reviewer 1 mentions that the suggested reference related to Omega 3 and PD was not included. This is not essential if you have provided an alternative citation to support this statement, but please do check that all statements are supported by relevant papers.

Please reupload your revised manuscript without tracked changes as the main "Manuscript" file, removing the original file to avoid confusion during peer review.

We look forward to receiving your revised manuscript.

Kind regards,

Sarah Jose, Ph.D.

Staff Editor

Journal Requirements:

Additional Editor Comments (if provided):

Reviewers' comments:

Reviewer's Responses to Questions

**Comments to the Author**

Reviewer #1: (No Response)

Reviewer #2: All comments have been addressed

publication criteria ? Is the manuscript technically sound, and do the data support the conclusions? The manuscript must describe methodologically and ethically rigorous research with conclusions that are appropriately drawn based on the data presented.

Reviewer #1: Yes

Reviewer #2: Yes

3. Has the statistical analysis been performed appropriately and rigorously?

Reviewer #1: Yes

Reviewer #2: Yes

4. Have the authors made all data underlying the findings in their manuscript fully available (please refer to the Data Availability Statement at the start of the manuscript PDF file)?

Reviewer #1: Yes

Reviewer #2: Yes

5. Is the manuscript presented in an intelligible fashion and written in standard English?

Reviewer #1: Yes

Reviewer #2: Yes

Reviewer #1: Dear Author,

You have made substantial and thorough revisions in response to likely reviewer comments. The manuscript is now clearer, more methodologically transparent, and better contextualized. Key issues have been addressed comprehensively.

One of my suggestions have not been considered: I suggested that to use the references for each nutritional items that you have listed. In Omega-3 fatty acid cite this article: Effect of Clupeonella grimmi (anchovy/kilka) fish oil on dysmenorrhea https://doi.org/10.26719/2010.16.4.408. It is a valuable reference and relevant to the introduction section. Please add the same reference.

Reviewer #2: It is accepted, given that the authors have made all the corrections.

(what does this mean? ). If published, this will include your full peer review and any attached files.). If published, this will include your full peer review and any attached files.

**Do you want your identity to be public for this peer review?** For information about this choice, including consent withdrawal, please see our Privacy Policy .

Reviewer #1: No

Reviewer #2: **Yes:** Zohreh KarimiankakolakiZohreh Karimiankakolaki

---

## [Decision Letter · Decision Letter 2]

9 Feb 2026

PGPH-D-25-03086R2

Prevalence of Primary Dysmenorrhea and its associated factors among adolescent girls in rural setting of Pakistan

Dear Dr. Das,

Thank you for submitting your manuscript to PLOS Global Public Health. After careful consideration, we feel that it has merit but does not fully meet PLOS Global Public Health’s publication criteria as it currently stands. Therefore, we invite you to submit a revised version of the manuscript that addresses the points raised during the review process.

The manuscript has been evaluated by one reviewer, and their comments are available below.

The reviewer has raised concerns that need attention. They raise conceptual, methodological, and interpretative issues and find that the manuscript requires further clarification and strengthening to enhance the scientific rigor and transparency of the study.

Could you please revise the manuscript to carefully address the concerns raised?

We look forward to receiving your revised manuscript.

Kind regards,

Katrien G. Janin, PhD

Staff Editor

Journal Requirements:

Additional Editor Comments (if provided):

Reviewers' comments:

Reviewer's Responses to Questions

**Comments to the Author**

Reviewer #3: (No Response)

publication criteria ? Is the manuscript technically sound, and do the data support the conclusions? The manuscript must describe methodologically and ethically rigorous research with conclusions that are appropriately drawn based on the data presented.

Reviewer #3: Yes

3. Has the statistical analysis been performed appropriately and rigorously?

Reviewer #3: Yes

4. Have the authors made all data underlying the findings in their manuscript fully available (please refer to the Data Availability Statement at the start of the manuscript PDF file)?

Reviewer #3: Yes

5. Is the manuscript presented in an intelligible fashion and written in standard English?

Reviewer #3: Yes

Reviewer #3: This manuscript addresses an important and underexplored public health issue by examining the prevalence of primary dysmenorrhea and its associated factors among adolescent girls in a rural district of Pakistan. The focus on a rural population, combined with a community-based cluster sampling design and inclusion of both in- and out-of-school adolescents, represents a notable strength and adds potential value to the existing literature.

Nevertheless, several conceptual, methodological, and interpretative issues require further clarification and strengthening to enhance the scientific rigor and transparency of the study. The following comments are provided to assist the authors in improving the justification of the research context, clarifying key methodological decisions, and ensuring that conclusions are appropriately supported by the data.

Lines 57–62

The manuscript would benefit from a clearer articulation of the specific characteristics of rural Pakistan that justify a separate investigation. Evidence from urban settings in Pakistan may not be generalizable to rural contexts due to substantial differences in socioeconomic conditions, educational access, healthcare availability, cultural norms, and dietary patterns. Strengthening this paragraph by explicitly explaining why findings from urban populations cannot be extrapolated to rural adolescents would improve the study rationale and contextual relevance.

Lines 63–72

The discussion on BMI and dysmenorrhea should be strengthened by explicitly acknowledging the existing inconsistency in the literature. Previous studies report conflicting associations between BMI and dysmenorrhea (protective, harmful, or null), and this heterogeneity warrants clearer discussion. Highlighting these inconsistencies would better justify the need for further investigation and contextualize the current findings within the broader evidence base.

Lines 49–56

This paragraph currently provides descriptive definitions of primary and secondary dysmenorrhea but contributes limited conceptual value. If retained, it should be strengthened by explicitly discussing the physical, psychological, educational, and social consequences of both PD and SD among adolescent girls. Emphasizing impacts such as school absenteeism, reduced quality of life, and psychosocial stress would better establish the urgency and public health relevance of studying dysmenorrhea.

Lines 66–68

The manuscript introduces omega-3 fatty acids, magnesium, vitamin B6, calcium, and iron as relevant nutritional factors, but the empirical or mechanistic basis for their inclusion is not sufficiently explained. The authors should briefly summarize existing evidence or biological mechanisms linking these micronutrients to dysmenorrhea (e.g., inflammation, prostaglandin synthesis, neuromuscular function). In addition, it would strengthen the argument to explain why these nutrients are likely to be deficient in rural settings, including discussion of limited dietary diversity, reliance on staple-based diets, low animal-source food consumption, and constrained food access.

Lines 68–70

The statement suggesting reduced physical activity influenced by urbanization requires clarification. Many studies report higher levels of physical activity in rural populations due to domestic and agricultural labor. The authors should clarify whether urbanization in rural Pakistan refers to lifestyle transitions (e.g., increased screen time, reduced recreational activity, schooling patterns) and support this claim with references or contextual explanation.

Line 145

The manuscript states that primary dysmenorrhea was defined as menstrual pain “without underlying reproductive disease,” but it is unclear how underlying gynecological conditions were excluded. Please clarify whether this was based solely on self-report, symptom screening, medical history, or any clinical assessment, and discuss the implications of potential misclassification.

Line 168

Please clarify how sleep duration was assessed. Was sleep measured using a validated questionnaire, structured interview, or a single self-reported question? If self-report was used, this limitation should be acknowledged.

Lines 169–171

It is unclear whether a validated physical activity instrument was used or whether activity levels were assessed through simple recall questions. Please specify the instrument, scoring method, and rationale for the chosen cut-off (≥180 minutes/week).

Lines 198–200

Further clarification is needed on how the multiple imputation process was validated. Please indicate which variables were included in the imputation model, how convergence was assessed, and whether sensitivity analyses were conducted to ensure robustness.

Line 206

The use of a significance threshold of p < 0.1 in multivariable analysis is uncommon in population-based epidemiological studies. A clear methodological justification is required, including whether this threshold was chosen due to exploratory objectives, sample size considerations, or concerns about statistical power.

Lines 264–267

The authors report that each 10 g increase in carbohydrate intake slightly increased the odds of PD. It would be informative to clarify whether this association differs by BMI category (underweight, normal weight, overweight). Additional stratified analysis or discussion would help interpret whether this finding applies across all nutritional statuses or specific subgroups.

Lines 296–297

Caution is warranted when describing overweight/obesity and cloth pad reuse as “protective” factors. These statements should be immediately accompanied by nuanced discussion and clear acknowledgment of alternative explanations, residual confounding, and potential misinterpretation, particularly given the public health implications. Integrating the interpretation directly with the results presentation would improve clarity and reduce the risk of overstatement.

Lines 298–305

The explanation of birth order effects relies heavily on speculative cultural interpretations. While the proposed psychosocial mechanisms are interesting, they are not sufficiently supported by empirical data. The authors should either provide supporting references or clearly tone down speculative language and present these explanations as hypotheses requiring further investigation.

(what does this mean? ). If published, this will include your full peer review and any attached files.). If published, this will include your full peer review and any attached files.

**Do you want your identity to be public for this peer review?** For information about this choice, including consent withdrawal, please see our Privacy Policy .

Reviewer #3: No

---

## [Decision Letter · Decision Letter 3]

9 Mar 2026

Prevalence of Primary Dysmenorrhea and its associated factors among adolescent girls in rural setting of Pakistan

PGPH-D-25-03086R3

Dear Dr. Das,

We are pleased to inform you that your manuscript 'Prevalence of Primary Dysmenorrhea and its associated factors among adolescent girls in rural setting of Pakistan' has been provisionally accepted for publication in PLOS Global Public Health.

Best regards,

Julia Robinson

Executive Editor

Reviewer Comments (if any, and for reference):

Reviewer's Responses to Questions

**Comments to the Author**

Reviewer #3: (No Response)

publication criteria ? Is the manuscript technically sound, and do the data support the conclusions? The manuscript must describe methodologically and ethically rigorous research with conclusions that are appropriately drawn based on the data presented.

Reviewer #3: Yes

3. Has the statistical analysis been performed appropriately and rigorously?

Reviewer #3: Yes

4. Have the authors made all data underlying the findings in their manuscript fully available (please refer to the Data Availability Statement at the start of the manuscript PDF file)?

Reviewer #3: Yes

5. Is the manuscript presented in an intelligible fashion and written in standard English?

Reviewer #3: Yes

Reviewer #3: Dear authors,

Thank you to the authors for addressing the review comments. Most points have been appropriately revised; however, one item still requires emphasis. In the response to Lines 57–62, the authors should explicitly describe the characteristics of rural areas that distinguish them from urban settings, particularly in rural Pakistan.

Thank you.

(what does this mean? ). If published, this will include your full peer review and any attached files.). If published, this will include your full peer review and any attached files.

**Do you want your identity to be public for this peer review?** For information about this choice, including consent withdrawal, please see our Privacy Policy .

Reviewer #3: No
